# Hrip1 Induces Systemic Resistance against Bean Aphid (*Megoura japonica* Matsumura) in Common Beans (*Phaseolus vulgaris* L.)

**DOI:** 10.3390/microorganisms10061080

**Published:** 2022-05-24

**Authors:** Khadija Javed, Yong Wang, Humayun Javed, Talha Humayun, Ayesha Humayun

**Affiliations:** 1Department of Plant Pathology, Agriculture College, Guizhou University, Guiyang 550025, China; 2Department of Mountain Agriculture and Environmental Sciences, Kohsar University Murree, Murree 47150, Pakistan; 3Institute for Biological Control Julius Kühn-Institut (JKI), 64287 Darmstadt, Germany; 4Rothamsted Research West Common Harpenden, Hertfordshire AL5 2JQ, UK; hjhumayun@gmail.com; 5Department of Surgery, Federal Government Polyclinic Hospital (P.G.M.I.), Islamabad 04403, Pakistan; talhahumayun7@gmail.com; 6Department of Surgery, Rawalpindi Medical University, Rawalpindi 46000, Pakistan; 7Department of Clinical Studies, Pir Mehr Ali Shah-Arid Agriculture University, Rawalpindi 46300, Pakistan; ayeshahumayun221@yahoo.com

**Keywords:** *Alternaria tenuissima*, *Megoura japonica*, *Phaseolus vulgaris*, nymphal instars, fecundity, expressions of defense-related genes

## Abstract

The emerging elicitor protein Hrip1 was evaluated for sublethal effects and biocontrol potential in the common bean *Phaseolus vulgaris*. In *Megoura japonica* Matsumura, purified elicitor protein Hrip1 was investigated for impacts on endurance, life expectancy, juvenile expansion, fully grown procreative performance, and pathogen–pest interface. The multi-acting entomopathogenic effects of the active compounds of *Alternaria tenuissima* active on Hrip1 in common bean (*Phaseolus vulgaris* L.) plants were also investigated. *Megoura japonica* population expansion was reduced by Hrip1 treatments (second and third generations). In a host selection test, control plants colonized quicker than Hrip1-treated *P. vulgaris* plants. Hrip1 influenced the longevity, development, and fertility of insects. Hrip1-elicitor protein concentrations aided *M. japonica* nymph development. Similarly, seedlings treated with Hrip1 generated fewer offspring than seedlings not treated with Hrip1. Hrip1 altered plant height and leaf surface structure, reducing *M. japonica* reproduction and colonization. Hrip1-treated *P. vulgaris* seedlings exhibited somewhat increased amounts of jasmonic acid, salicylic acid, and ethylene (ET). The integrated management of insect pests and biocontrol with Hrip1 in the agroecosystem appears to be suitable against *M. japonica* based on these findings.

## 1. Introduction

Pathogens that are successful must be able to recognize and overcome host-plant defenses. In response to pathogens that evade, tolerate, or decrease basal defenses, plants have evolved resistance (R) proteins, resulting in gene-for-gene resistance. Plants have a combination of inducible and constitutive defensive systems that help them resist illness. [1]. Researchers use conserved, essential chemicals to name microorganisms and pathogens and pattern-recognition receptors in innate immunity recognize them (PRR) [2]. They trigger oxidative bursts, produce nitric oxide (NO), secondary metabolites, and HR by raising extracellular pH [1,2]. Cells surrounding the infection site initiate these resistance responses. As a result, the plant acquires systemic acquired resistance (SAR) against a variety of diseases [3]. Plants have two defense systems built in; flagellin, a pathogen-deterrent molecular pattern (MAMPs or PAMPs), for example, assists plants in identifying bacteria and pathogens. PAMP-induced immunity (PTI) refers to the innate immunity of plants which is triggered by PAMP via numerous plant transmembrane pattern-recognition receptors (PRRs) [4]. Gene-for-gene resistance is a sort of defense that primarily occurs within plants. The pathogen-secreted elicitors are likened to R proteins in this case [5]. R proteins activate hypersensitive reactions, oxidative stress, NO generation, extracellular pH elevation, cell wall augmentation, and pathogen-related protein expression as part of effector-triggered immunity (ETI) [3,5]. This type of reaction begins at the infection site and spreads to neighboring cells that are not infected, improving the plant’s ability to combat infections [6].

Elicitors are responsible for stimulating the defense response and mechanism of action in plants in both biotic and abiotic ways [7]. Numerous elicitors have been identified from bacteria, viruses, oomycetes, and fungus, among other organisms. Proteins, peptides, glycoproteins, lipids, and oligosaccharides have all been used as elicitor molecules, and some of these elicitors have even been employed to help plants resist pathogens [8,9]. Ion inflow is frequently related to hypersensitive response (HR) and reactive oxygen species (ROS) such H_2_O_2_ and O_2_. These compounds function as signaling elicitors [10]. Elicitors are plant defense groups that influence both host and non-host plants and are categorized as race-specific or universal [11]. Some chemical pesticides may be replaced by elicitors to assure food safety [12,13,14,15,16,17,18]. Aphid defense responses have been studied in a number of aphid–plant interactions. Aphid resistance in *Brassica napus* (Brassicaceae) dropped immature plant endurance and population growth of immature *Plutella xylostella* (Lepidoptera: Plutellidae) [19]. Jasmonic acid (JA), salicylic acid (SA), and ethylene (ET) all excite defensive responses in plants [20,21,22]. Several studies have shown that JA and SA play a role in causing an aphid response by increasing the expression of genes such as *LOX1* (lipoxygenase) and *PAL* (phenylalanine ammonia-lyase) after aphids feed on them [23].

*Alternaria* spp. is a source of elicitors. When given a strain of *Alternaria alternata*, tobacco BY-2 cells produced chitinases. Additionally, *A. tenuissima* protein helped cotton grow and improved defense-related enzyme work [24,25]. When recombinant Hrip1 from the necrotrophic fungus *A. tenuissima* was administered to the host plants, local and systemic defense responses were found [25,26]. The JA and SA pathways make plants more resistant to it. The current study examined Hrip1, an elicitor protein produced by the necrotrophic fungus *A. tenuissima*, and its impact on *M. japonica* management [26]. These findings provide insight into the function of Hrip1, how it influences the biological control of *M. japonica*, and what these implications indicate for pest management in the future. The aim of this study was to look into the activity and molecular mechanism of the *A. tenuissima* -derived elicitor protein Hrip1 in the induction of bean aphid resistance in common bean plants. The impacts of Hrip1 on *M. japonica* control, as well as the roles and mechanisms of PeaT1 and PeBC1 on *M. japonica* control, were investigated in this work to analyze the prospective influence of Hrip1 on *M. japonica*. Trichomes were discovered on the leaf surface structure, thus prompting researchers to examine the contents of the JA and SA gene expression from JA and SA. This research also includes information on Hrip1 function, mechanism, and the effects of the integrated management of the bean aphid (*M. japonica*).

## 2. Materials and Methods

### 2.1. Insect and Plant Colonies

The focus of this research was to grow bean aphid (*M. japonica* Matsumura) and common bean (*Phaseolus vulgaris* L.) colonies in a controlled growing season before the tests. *Megoura japonica* was found nearby *Brassica oleracea*. Then it was transferred to seedlings of common bean (*Phaseolus vulgaris* L., Guizhou cultivar F1870). During the experiment, a colony of *M. japonica* was kept at room temperature for 6 months before the experiment. For 20–40 s, the seeds of *P. vulgaris* were cleaned in 75% ethanol. Then washed with water and soaked for 2–3 d prior to use.

### 2.2. Hrip1 Expression, Purification and Evaluation

The Hrip 1 protein elicitor gene was expressed using yeast peptone dextrose (YPD). Hrip 1 was grown in 25 mL of liquid YPD medium containing 1% dextrose, 0.5% yeast extract, and 1% peptone. Yeast peptone dextrose YPD medium was agitated at 200 rpm at 30 °C overnight before being transferred to 1 mL liquid medium of BMGY (Millipore, Crop., Billerica, MA, USA) with 100 mM KH_2_PO_4_ (pH = 7.0). The medium was shaken at 200 rpm in a shaker until its absorbance reached 600nm. The pellet was obtained by centrifuging the medium for 10 min at 25 °C at 5000 rpm. The pellet was resuspended in 100 mL of buffered methanol-complex medium (BMMY) supplemented with 1.3 g yeast and incubated for 72 h at 29 °C in a shaker at 200 rpm. The protein supernatant was filtered via a 0.22 μm membrane pore size syringe filter. A His-Tag Purification column was used for further purification (GE Healthcare, Waukesha, WI, USA). To elute the protein elicitor, three buffers were used: Buffer A (50 mM Tris-HCl + 200 mM NaCl) to remove contaminants and bind the proteins in the columns, Buffer B (50 Mm Tris-HCl + 200 Mm NaCl + 20 Mm Imidazole) to balance the columns, and Buffer C (50 mM Tris-HCl + 200 mM NaCl + 500 mM imidazole); after that, the protein was centrifuged in a desalting tube. To remove the concentrated salt, the desalting columns were washed three times using a buffer (50 mM Tris-HCl, pH 8.0). After desalting and centrifuging the protein, it was washed three times with 50 mM Tris-HCl, pH 8.0. Protein fraction trituration with buffer and desalting tube (50mM Tris-HCl, pH 8.0) was performed. A protein marker (Thermo Scientific, Rockford, IL, USA) was used to quantify the Hrip1 elicitor protein (Thermo Scientific) [27].

### 2.3. Megoura japonica Infestation on the Plant

This experiment was designed to estimate the size of the *M. japonica* population that had settled. *Phaseolus vulgaris* were soaked in 72.46 μg mL^−1^ Hrip1 for one day. Four organic seeds were developed (Flora Guard substrate). After seven d, *Phaseolus vulgaris* seedlings were sprayed with 72.46 μg mL^−1^ Hrip1 solution and inoculated with 15 *M. japonica* adults. Seedlings were treated weekly. Every five days after inoculation, aphids were counted. The data fractions were used to analyze and comprehend the data. Controls and negative controls were treated with water and 72.46 μg mL^−1^ of a buffer (50 mM Tris-HCl, pH 8.0), respectively. Plants were caged in transparent, breathable mesh. Each time, four replicates were used.

### 2.4. Growth Rate of M. japonica

The purpose of this experiment was to see if feeding Hrip1-treated or control seedlings enhanced the intrinsic growth of *M. japonica*. *Phaseolus vulgaris* seeds were managed as in (Section 2.3). Seeds were splattered with 72.46 μg mL^−1^ of Hrip1 pure protein solution after a day. Altogether, a glass tube was lined with cotton gauze–isolated sprouts, and aphid mobility was restricted on the leaf in a plastic ecological cage (2.7 × 2.7 × 2.7 cm). To prevent mechanical damage to the leaf, the perimeter of the ecological cage was sponge-coated. Every 12 h, the *M. japonica* instar was checked for nymph production. In order to avoid crowding, newly molted nymphs were counted two times every day to govern the total aggregate of time and offspring created. This was done again five days later on seeds and plants. The 30 duplicates of each treatment were used in the experiment. The intrinsic rate was calculated as follows:*r_m_* = 0.738 × (ln *Md*)/*Td*

*Md* counts the set of newborn nymphs in a *Td* development phase (the period of time among an aphid’s infancy at first reproduction)

### 2.5. Megoura japonica Bioassay

This study’s goal was to examine *M. japonica* nymph development and fertility. On *P. vulgaris* plants, Hrip1 was tested against *M. japonica* at 72.46, 43.47, 21.73, and 18.10 μg mL^−1^. This was calculated using the Bradford assay. Using a separate spray bottle, approximately 3 mL of Hrip1 was sprayed into the *M. japonica* plants at the three-leaf stage. Water and buffer were used to treat the controls (50 mM Tris-HCl, pH 8.0). Then, 3–6 *M. japonica* (0–6 h old) were given to *P. vulgaris* plants and desiccated instantaneously. The entire amount of descendants formed by all aphid instars was used to calculate the overall nymph development period, while aphid longevity was derived from the number of days they lived. The bioassays were performed in triplicate at three different temperatures (20, 23, and 26 °C).

### 2.6. Hrip1 Impacts on M. japonica Development and Structure

The goal of this study was to see how Hrip1 affected *M. japonica* growth and structure. *Phaseolus vulgaris* seeds and seedlings were treated as described in Section 2.3. A 3.5 percent glutaraldehyde solution in 0.1 M phosphate solution was used to collect samples for up to two days (pH 7.2). In total, five 2-hourh submersions in 1% osmic acid were performed on all samples. It was used for 15 min with an ethanol gradient of 100% to 95% to 90% to 80% to 70% to 30%. An EM critical point drier Leica dried all critical points (CPD030; Leica Bio-systems, Wetzlar, Germany). Altogether, samples were inspected with a Hitachi H-7650 TEM. Hrip1-treated colonies were measured in 10 duplicates.

### 2.7. HPLC/MS

The goal of this study was to quantify the amount of SA, JA, and ET accumulated in this way; seeds and seedlings were handled as described previously. Seedlings’ aerobatic sections were collected for SA, JA, and ET [28]. This was done using an HPLC/MS (Shimazu Research Instruments, ODS-C18, 3 m, and 2.1 per 150 mm Kyoto, Japan). Methanol mobile phase, 60%, and 4 °C sample temperature were used during HPLC. The Sim system was set in negative ion mode with the following parameters: solvent 250 °C, heat block 200 °C, gas flow rates 10 L/min, nebulizing gas 1.5 L/min, detector voltage 1.30 kV, interface 3 kV (SA m/z: 137.00; JA: 209.05).

### 2.8. Gene Expressions

TransGen Biotech (Beijing, China) kits were used to extract RNA, synthesize cDNA, and perform Q-RT-PCR (ABI 7500 Real-Time PCR System). The RNA was tested using an NP80 nano-photometer. PHAVU_002G175500g, PHAVU_001G017800g, PHAVU_003G111500g, PHAVU_001G000800g, PHAVU_001G001300g, PHAVU_002G06700g, PHAVU_003G096400g, PHAVU_003G011600g, PHAVU_006G048600, PHAVU_008G057700, PHAVU_008G272800, PHAVU_011G176100, and PHAVU_011G17200 were tested for the JA, SA, and ET pathway. Internal reference was *β-actin* gene [29]. Table 1 lists the primers used. The relative fold expression of genes was assessed using 2^−ΔΔCT^ method [30].

### 2.9. Data Analysis

Using Statistix software version 8.1, (Analytical Software, Tallahassee, FL, USA) ANOVA and Levene’s tests compared two treatments, while LSD and ANOVA compared three or more treatments (Tallahassee, FL, USA). This data was first square-root transformed before analysis. To eliminate disparities, authors used a 95 percent probability LSD test on treatment variables such as Hrip1 elicitor concentrations and temperature regimes for gene expression the comparative CT (2^−∆∆CT^) method was used, and the fold changes having protein elicitor and buffer applied were compared at (α = 0.05)

## 3. Results

### 3.1. M. japonica Indoors

Hrip1 induced *M. japonica* resistance in two ways. *M. japonica*–treated *P. vulgaris* seedlings had significantly reduced aphid populations. Figure 1 compares the population declines in the Hrip1 treatment to the buffer and control. Aphid development was extended more in the Hrip1 treatment than control; however, the everyday reproductive capacity of *M. japonica* was reduced when they were fed Hrip-treated seedlings (second and third nymphal instars). Both the second and third generations showed lower growth rates, and when *M. japonica* was fed Hrip1-treated seedlings, all generations showed lower growth rates, according to the findings (Figure 2).

### 3.2. Hrip1 Influenced M. japonica Nymphal Development and Fecundity

The interaction of different Hrip1 concentrations with three temperature regimes influenced *M. japonica’s* overall development period. As Hrip1 concentrations increased, so did the nymphal instar development time (Figure 3).

For the fourth nymphal instar development time of 3.9 d at 72.46 µg mL^−1^ and 20 °C, a concentration of 18.10 µg mL^−1^ at 26 °C produced a minimum 1.7 d nymph growth. Aphid fecundity influenced Hrip1 concentrations and temperatures (Figure 4). The experiment found that fecundity was lowest at 26 °C and highest at 23°C.

### 3.3. Hrip1 Influenced the Development and Structure of P. vulgaris

Hrip1 significantly influenced the plant height and surface structure of *P. vulgaris* seedlings compared to control seedlings (Figure 5). Hrip1-treated seedlings had significantly more trichomes than control seedlings, with 75.10 ± 0.21 mm^−2^ in Hrip1-treated seedlings versus 24.34 ± 0.11 mm^−2^ in control seedlings; *p* = 0.05. With a better surface environment and a more complex wax structure, aphid colonization should be more difficult.

### 3.4. SA, JA, and ET Quantities

JA, SA, and ET examine links among cuticular wax deposition, trichome number, and aphid infestation in Hrip1. JA, SA, and ET in Hrip1 seedlings were found to be higher (Figure 6). The development of aphid resistance in *P. vulgaris* required all three signaling pathways. The protein elicitor elicited an innate immunological or defensive response in *P. vulgaris* plants, according to the findings.

### 3.5. Defense-Related Gene Expression Fold Change

Hrip1 boosted defenses in *P. vulgaris* seedlings. Hrip1 treatment slightly upregulated all JA and SA pathway test genes (Figure 7 and Figure 8). The Log2 of all test genes was calculated using fold-change expression values, indicating that transcription triggered aphid confrontation.

## 4. Discussion

A new biological tool for pest control, elicitors, share a role in signaling system and plant defense [13,14,15,16,17,31]. PAMPs and MAMPs are abundant in both necrotrophic or biotrophic pathogenic bacteria and fungi [32]. Hrip1, an *A. tenuissima* protein elicitor, has antimicrobial and biocontrol potential against *M. japonica*. In the *P. vulgaris* crop, chemical elicitors like methyl-jasmonate and benzo-thiadiazole, as well as proteinase inhibitors, have been shown to significantly reduce herbivore pest activity. According to previous research, methyl salicylate reduces the population of *A. glycines* by up to 40% [32,33]. In this study, Hrip1 inhibited herbivores by altering physical plant characteristics. The results of this study are in line with the previous work that trichomes are the first step in building physical resistance to pathogens and herbivores. These affect herbivore shape and trichome density in *Solanum* spp. [34,35]. This study confirmed that Hrip1 reduced disease severity by initiating a photosynthetic process and increased induced resistance in *P. vulgaris* seedlings treated with Hrip1, which is in line with the previous work [36,37]. The lignin concentration of *Chrysanthemum indicum* increased aphid tolerance [38]. Our results are in agreement with previous studies which indicate that, in response to biotic and abiotic stress, plants produce trichomes and wax [39,40,41]. The results are in agreement with the previous work reporting that the Hrip1 elicitor had a negative influence on aphid fecundity. Hrip1-treated plants had far fewer aphids than the controls. Exogenous SA and MJ deplete aphid mean fecundity to 50% [32,41]. The findings of this study corroborate the work by Javed et al. that the lowest aphid fecundity was observed at lower temperatures, e.g., 20 °C, while the highest fecundity was observed at 26 °C, attributed to a declining proportion of metabolic activities [42]. The maximum nymphal development time was observed even at a lower temperature (20 °C), indicating that a one-degree temperature increase had an effect on the insect life cycle [43].

After being exposed to Hrip1, *P. vulgaris* became more resistant to *M. japonica* than before and a systemic defense response was triggered by beneficial bacteria; this response is controlled by a signaling system that links the plant hormones SA, JA, and ET, and it is controlled by the bacteria (ET). These results were in accordance with previous studies [44,45]. To activate PR genes via interactions with TGA transcription factors, the authors found that Hrip1 increased JA, SA, and ET-responsive gene levels, and the results of this corroborate the previous work where the data showed that Hrip1-mediated systemic defensive responses in *P. vulgaris* are influenced by relative expression levels [46]. Discoveries from the present work also support the previous research by Chaerle et al. [47]; secondary metabolite accumulation can help plants fight infection by generating mechanical barriers to pathogen growth, which stimulates phenolic metabolism and lignin synthesis [48]. Systemic defenses are activated when an aphid infests plants [49]. These findings suggest that our work helps us better understand how Hrip1 obtained from *A. tenuissima* works in *P. vulgaris* against management of *M. japonica* [50].

## 5. Conclusions

Hrip1 made *P. vulgaris* more resistant to aphids, which made the second and third generations of *M. japonica* less fertile and more likely to have aphids. Mechanical defenses played a small role in the resistance characteristics. The surface structure of *P. vulgaris* leaves changed after Hrip1 was added to the plant. SA, JA, and ET, which are thought to be involved in systemic defense responses in *P. vulgaris* when Hrip1 is used, had significant increases in relative expression levels. Hrip1 also had a significant impact on *M. japonica* lifespan features in the lab, but additional investigation is needed to make it even more effective in the field. In our research, the authors found that Hrip1, with its significant enhancement of *P. vulgaris* L. resistance against *M. japonica,* can be used as a “vaccine” to protect plants of *P. vulgaris* against the pest *M. japonica*.

## Figures and Tables

**Figure 1 microorganisms-10-01080-f001:**
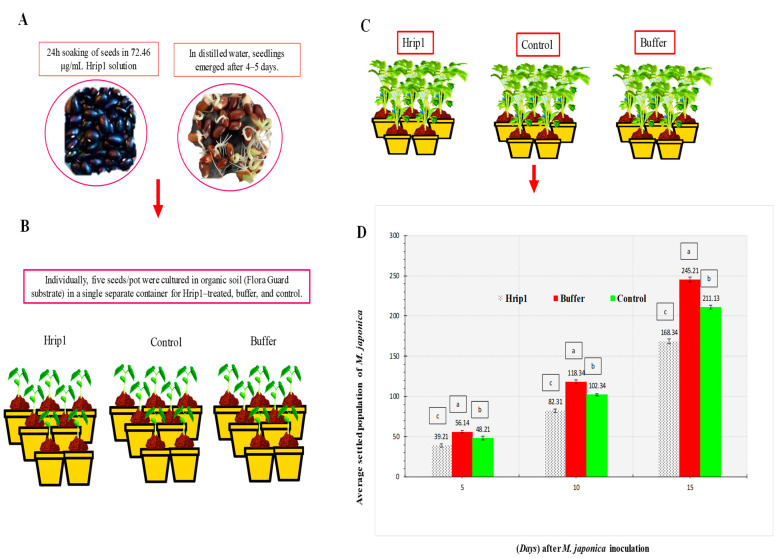
*Megoura japonica* population differences were observed in Hrip1-, control-, and buffer-treated *P. vulgaris* seedlings. Using one-way ANOVA and Levene’s test with SPSS 18.0, the LSD was at *p* = 0.05. (**A**,**B**) Treatment of seeds and seedling of *M. japonica*. (**C**) After 7 d, seedlings were inoculated with 15–20 adults of *M. japonica*; (**D**) seedlings treated with Hrip1 saw a substantial aphid population loss (mean ± SD); (a–c) the significant differences among Hrip1, buffer, and control. The study used a CRD, randomized statistical design; LSD and one-way ANOVA were used to compare data (*p* = 0.05).

**Figure 2 microorganisms-10-01080-f002:**
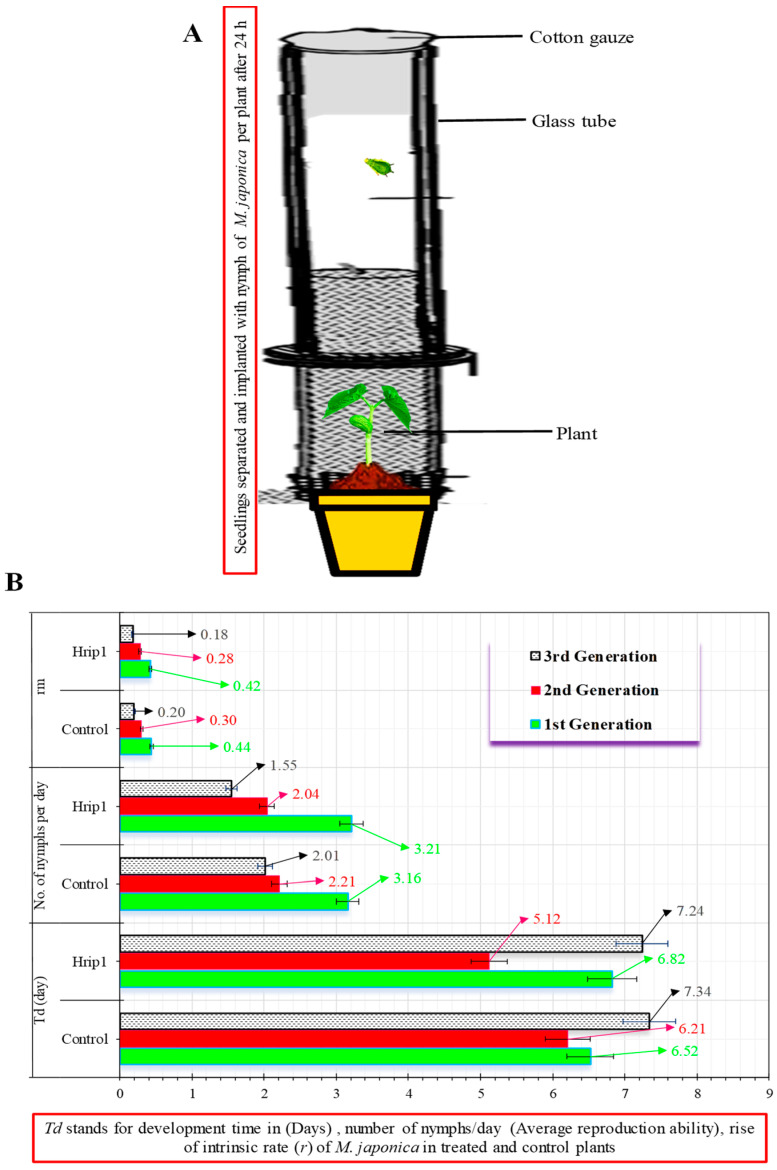
Time of the development, capacity of reproduction, and rate of the growth of *M. japonica* (**A**,**B**) in seedlings treated with Hrip1 and control (mean ± SD); CRD, designed in the study; SPSS 18.0 was used to compare data by LSD and one-way ANOVA at (*p* = 0.05).

**Figure 3 microorganisms-10-01080-f003:**
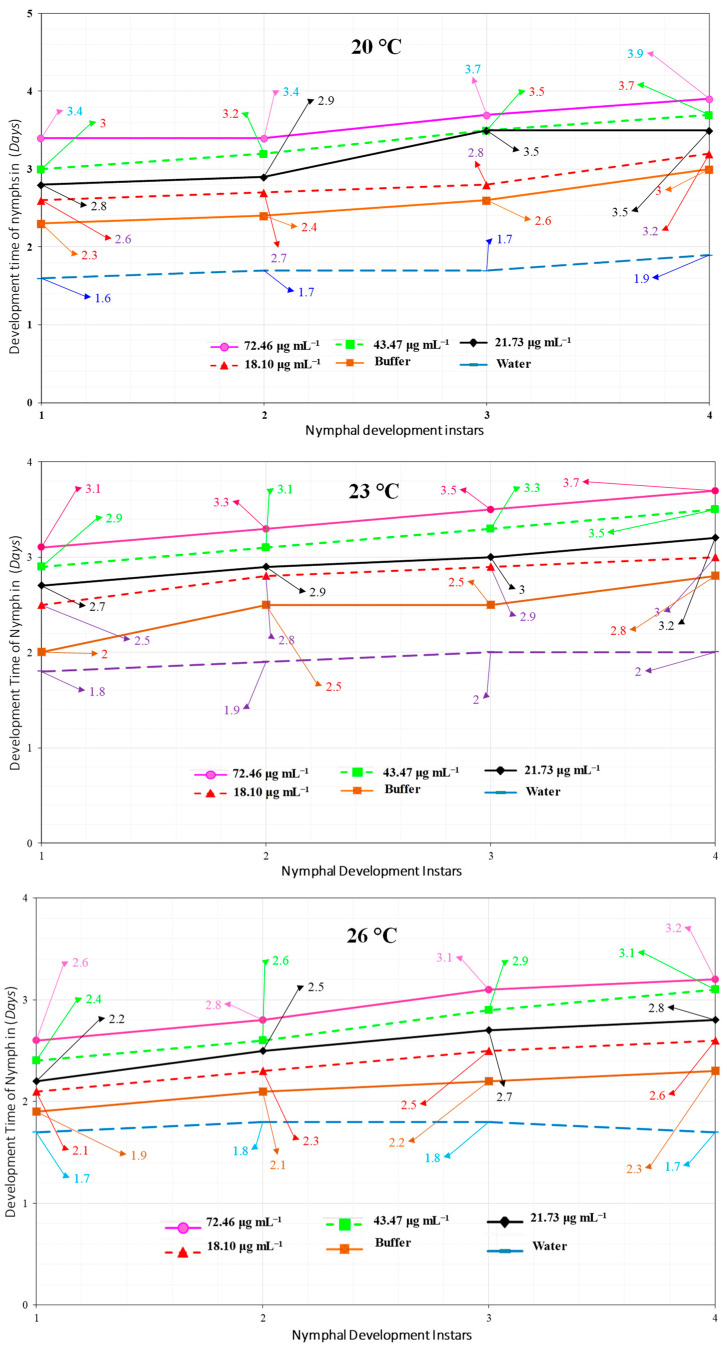
On *P. vulgaris* plants, the Hrip1 elicitor protein elicited the development of *M. japonica* nymphs at various doses and temperatures (20, 23, 26 °C) (*n* = 10; one-way ANOVA with factorial analysis; LSD at alpha = 0.05).

**Figure 4 microorganisms-10-01080-f004:**
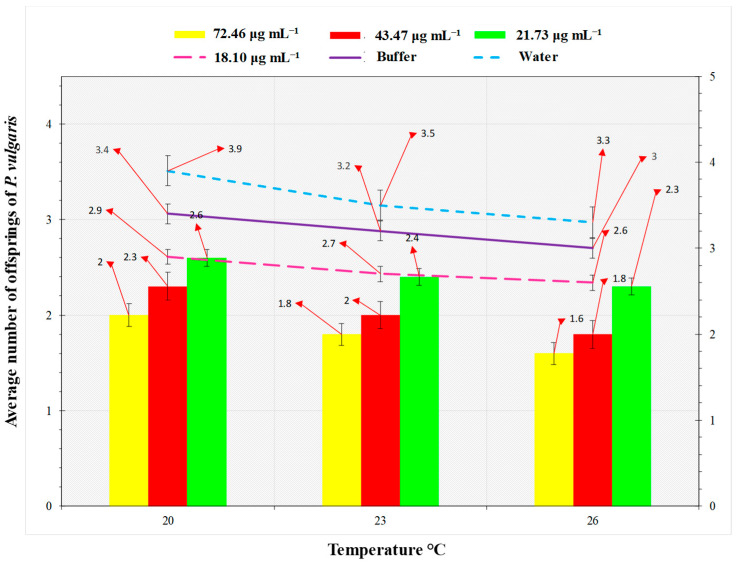
*Phaseolus vulgaris* average fecundity (*n* = 10). Fecundity lessened in *P. vulgaris* seedlings treated with Hrip1, (one-way ANOVA with factorial analysis; LSD at alpha 0.05).

**Figure 5 microorganisms-10-01080-f005:**
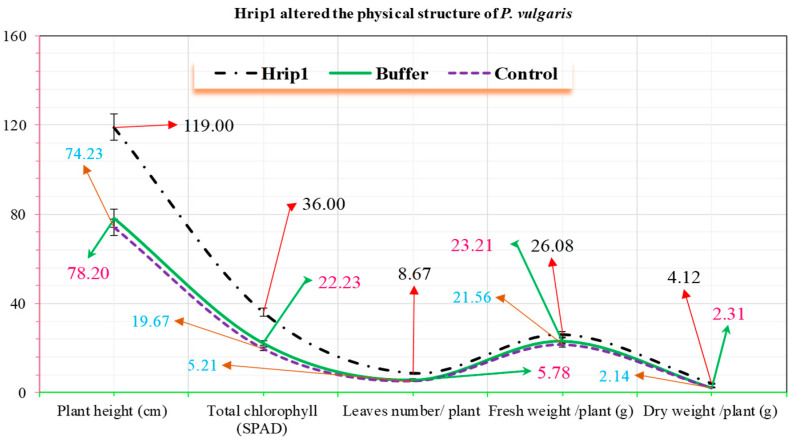
The effect of Hrip1 on the growth of treated and untreated seedlings (*n* = 10). Hrip1 and buffer seedling (mean ± SD) data were compared by LSD and one-way ANOVA with Levene’s test (*p* = 0.05) in SPSS 18.0.

**Figure 6 microorganisms-10-01080-f006:**
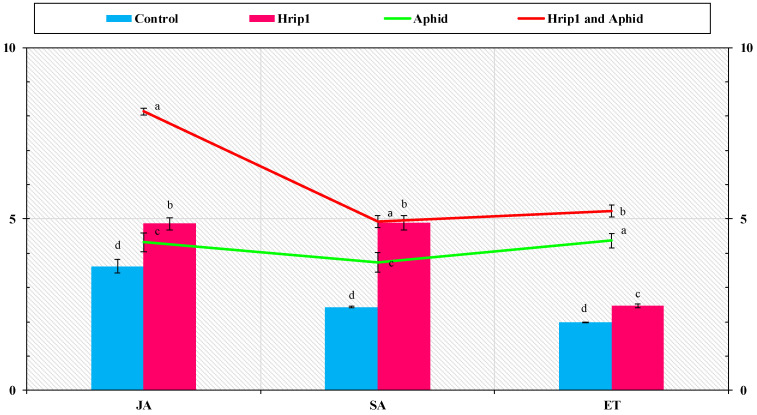
*Phaseolus vulgaris* seedling JA, SA, and ET levels (mean ± SD). An additional day after spraying, Hrip1 data were collected, and aphids were infected in both treatments. The LSD, ANOVA, and Leven’s test were used to compare data in 8.1 Statistix, with lower-case letters indicating significant JA, SA, or ET treatment differences. (*p* = 0.05).

**Figure 7 microorganisms-10-01080-f007:**
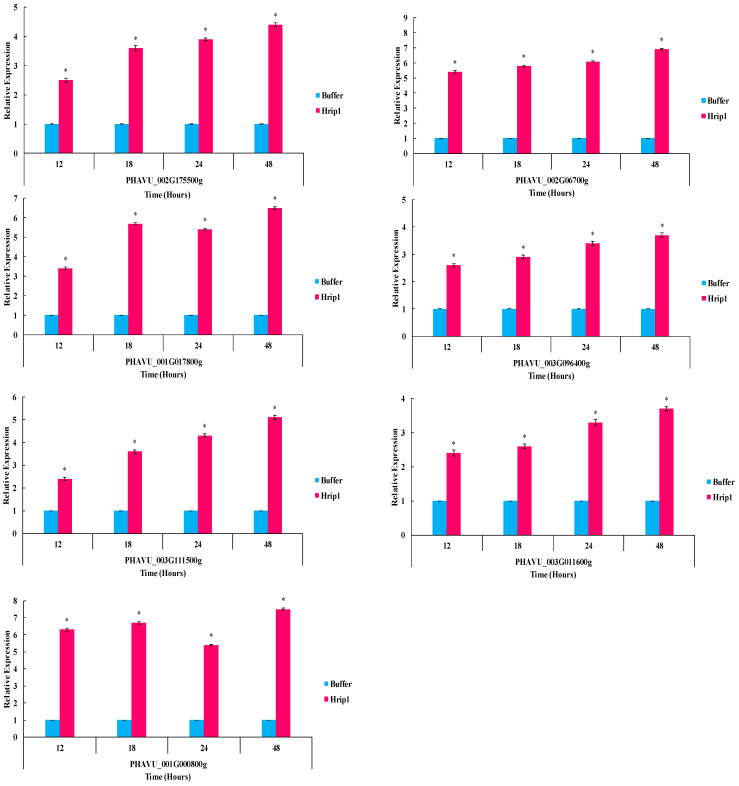
After Hrip1 elicitor treatment and aphid infestation, the relative expression of plant defense from the JA pathway was detected. An asterisk next to each gene indicates a significant difference from the buffer control, as determined using Student’s *t*-test (*p* < 0.05).

**Figure 8 microorganisms-10-01080-f008:**
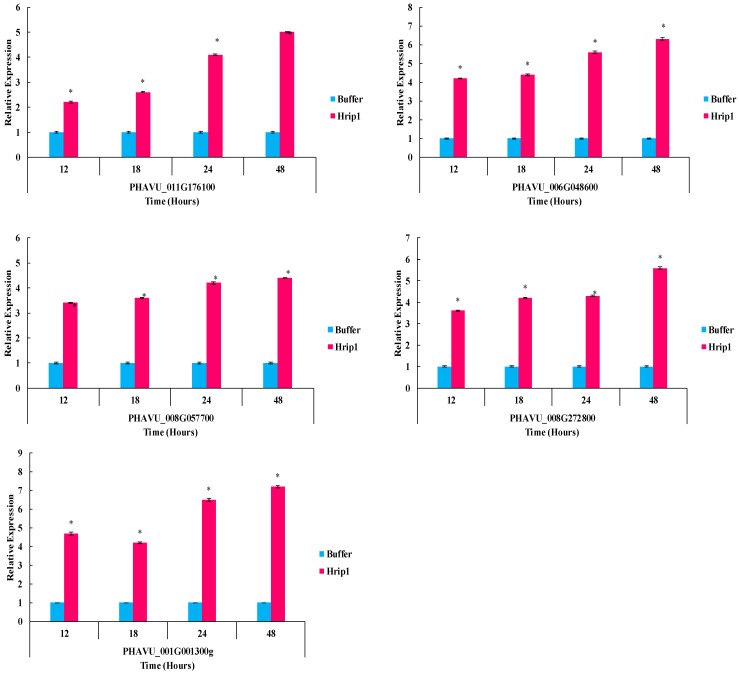
After Hrip1 elicitor treatment and aphid infestation, the relative expression of plant defense from the SA pathway was detected. An asterisk next to each gene indicates a significant difference from the buffer control (*p* < 0.05).

**Table 1 microorganisms-10-01080-t001:** Primers for plant defense genes JA, SA, and ET.

Test Genes	Forward Sequence (5′......3′)	Reverse Sequence (5′......3′)
PHAVU_002G175500g	GAAAAGCGTGGAAAGCTACG	AGCCATGAACGATGATCTCC
PHAVU_001G017800g	GGGAGAAGCTGCTGAAACAC	CCGACCTGAATATCGAAGGA
PHAVU_003G111500g	GAATTTCCCTGCTGCTCTTG	CTGGCTTAGCCTCAGGAATG
PHAVU_001G000800g	AGCCGCATGCTGTTCTCTAT	TTTTCATGAACAGCGCTCAC
PHAVU_001G001300g	TGAAATGGCCAAGAAGGAAC	GGCGACGAGACCGTATATGT
PHAVU_002G06700g	CTGATGAGCAGCAGCAGAAG	AAACGGGCATAAACAACAGC
PHAVU_003G096400g	ACGACCATGGGTTGCTAGTC	AATGCTTCAGCTTCCTTCCA
PHAVU_003G011600g	TAGTGATGGTGCAGGAGCTG	GATGCAAAGGCCTCATTGAT
PHAVU_006G048600	CAGGATGCTTGGGATGATCT	CAAGGGCCTTTCCTACTTCC
PHAVU_008G057700	TGCTTCACATGAATGGTGGT	CAACCCAAGTCTGCCACTTT
PHAVU_008G272800	TCCTTGTTGATGCCCACATA	CAAAGAAAAAGGGGGAGAGG
PHAVU_011G176100	CCCATGCACAGTGTACCAAG	ACCAATTAACCCCCAAGGAG
*β-actin*	GGAAAATCAGTCTCGGTTCAG	TCATACAGCAGCAAGCAC

## Data Availability

The required data set is already available in manuscript file; other data sets generated during the study are available upon request from corresponding author.

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
