# Peer review of "Hrip1 Induces Systemic Resistance against Bean Aphid (Megoura japonica Matsumura) in Common Beans (Phaseolus vulgaris L.)"

_microorganisms, 2022, doi:10.3390/microorganisms10061080_

Round 1
Reviewer 1 Report
Good evening dear authors,
I had the honor of being selected as a reviewer for your research paper which is focused on effect of purified fungal (Alternaria tenuissima) elicitor protein Hrip1 on resistance of Phaseolus vulgaris L. plant against Megoura japonica pest. As for the experimental part, your contribution article is well managed. However, several major and also minor deficiencies were found and they are listed below:
Fundamental deficiencies:
- In introduction, please define the aim of your research without any citations to be for potential readers clear the novelty statement.
- The discussion section does not include any your critical evaluation of the results obtained and rather reminds the continuation of the introduction. Please, use the references to confirm or refute your or previous claims dealing to your topic only.
- In the conclusion, it is stated that Hrip protein may possibly be cast-off as a "vaccine" to protect plants of vulgaris against the pest, M. japonica. Is it true that this study did not bring any positive observation like a significant increasing of plant (Phaseolus vulgaris L.) resistance to insects (M. japonica)?
- In article, it is stated that the surface structure of vulgaris leaves changed after Hrip1 was added to the plant. Please, can you provide any images?
Formal deficiencies:
- Unify please all references, so that the text of the cited article begins with a capital letter and the others are written in lower case.
- Unify please the chapter captions, all the initial letters of each word should be written in capital letters. Exceptions are prepositions, definite and indefinite articles.
- In title, words “induces and beans” should be written with capital letters.
- In line 55, molecule of oxygen should by written as O with 2 in lower index.
- Citations should be included in one break, for example [1,2,3], not [1][2][3].
- Minutes and seconds should be written as min and s only, respectively
- Divide the Figures so that the entire page is not blank. Next, enlarge the labels on the axes in the graphs to make them easier to read.
- In upper indexes, please use the minus sign symbol instead of the hyphen.
I believe that my comments will help to improve your contribution to be more attractive for potential readers.
With the best regards!
Your reviewer
Author Response
"Please see the attachment."

Reviewer 2 Report
Referees Comments
on the manuscript entitled " Hrip1 induces Systemic Resistance against Bean Aphid (Megoura japonica Matsumura) in Common beans (Phaseolus vulgaris L.)” for Microorganisms
The authors evaluated the effect of the elicitor protein Hrip1 produced by the fungus Alternaria tenuissima on the control of the bean aphid Megoura japonica. The topic of the article is interesting for the Microorganisms. However, the manuscript requires a significant revision.
Point 1: Many sentences in the Abstract and Introduction are too short and may be joined.
Point 2: Page 1, line 43-44: Check up the sentence with the acronym PAMP.
Point 3: Page 2, lines 48, 54, 58 - Links to References must be formatted in accordance with the rules of the journal.
Point 4: lines 52-54: Check up the sentences
Point 5: line 55: Give an explanation of the abbreviation HR.
Point 6: Page 2, line 77: Give the full name of the objects under study at the beginning of the section
Point 7: Page 2, lines 85 – 95: The description of the method is unclear.
Point 8: Page 3, lines 105-106: Control experiments with water and buffer concentrations are unclear
Point 9: Figure 1: In the caption to the figure, indicate what A. B, C, D means
Point 10: Page 5, line 171-173: Give a quantitative comparison of the data presented in Figure 1.
Point 11: Page 5: Give a detailed description of the results presented in Figure 2.
Point 12: Figures 3 and 4 duplicate each other.
Point 13: It is recommended to combine the Discussion section with the Results section.
Author Response
"Please see the attachment."

Round 2
Reviewer 1 Report
Good morning dear authors,
it can be concluded that your research article is sufficiently improved. Please, be consistent during the proofs, as many typos were still found, as follows:
a) gaps between units and physical quantities miss
b) units of physical quantities should not be written in italics
c) unify font size
I will recommend the acceptance of your article.
With the best regards!
Your reviewer
Reviewer 2 Report
The corrected version can be published without further editing.